

# Dosimetric comparison and evaluation of different convergence modes in nasopharyngeal carcinoma using VMAT treatment deliveries

Maoying Lan[1,*], Rui Wu[1,*], Guanhua Deng[2], Bo Yang[1], Yongdong Zhuang[1], Wei Yi[1], Wenwei Xu[1] and Jiancong Sun[1]

[1] Department of Radiation Oncology, The First Affiliated Hospital of Guangzhou Medical University, GuangZhou, GuangDong, China
[2] Department of Radiation Oncology, Guangdong Provincial People's Hospital (Guangdong Academy of Medical Sciences), Southern Medical University, GuangZhou, GuangDong, China
* These authors contributed equally to this work.

Corresponding authors
Wenwei Xu, wenweixu@126.com
Jiancong Sun, sunjiancong@126.com

## ABSTRACT

**Background:** This study investigates the impact of convergence mode (CM) in Eclipse (Varian Medical Systems) on the quality and complexity of volumetric modulated arc therapy (VMAT) plans for nasopharyngeal carcinoma (NPC) patients.

**Methods:** We retrospectively analyzed data from 21 NPC patients. For each patient, three VMAT plans with different CM settings (Off, On, and Extended) were created using identical optimization objectives. Plan quality was assessed using indices such as the conformity index (CI) and homogeneity index (HI), as well as evaluating target coverage and sparing of organs at risk (OARs). Complexity was measured by metrics including average leaf pair opening (ALPO), modulation complexity scores for VMAT (MCSv), monitor units (MUs), and optimization time. Dosimetric verification was performed based on the gamma pass rate.

**Results:** Different CM settings can generate treatment plans that meet clinical dose objectives for planning target volumes (PTVs) and OARs. The 'On' or 'Extended' CM settings improved CI and HI for the NPC target volume and reduced OAR doses, especially the mean dose, without compromising target coverage. The 'Extended' CM setting produced the most favorable outcomes. ALPO values for CM settings 'Off', 'On', and 'Extended' were 29.1 ± 4.3 mm, 28.6 ± 4.2 mm, and 28.4 ± 4. 2 mm, respectively. MCSv values for these settings were 0.1730 ± 0.0215, 0.1691 ± 0.0204, and 0.1693 ± 0.0208, respectively. MUs were 796.2 ± 110.8 for CM 'Off', 798.6 ± 106.1 for CM 'On', and 799.7 ± 103.6 for CM 'Extended', with no significant differences ($p > 0.05$). Gamma pass rates for all plans were above 99% (3%/3 and 2%/2 mm), with no significant differences among groups ($p > 0.05$). The average optimization times for CM settings 'Off', 'On', and 'Extended' were 14.4 ± 3.2, 35.9 ± 8.9, and 145.6 ± 50.3 min, respectively ($p < 0.001$).

**Conclusion:** CM usage can improve the CI and HI of the target volume and decrease the dose to OARs in VMAT plans for NPC patients. This study suggests that CM can be a valuable tool in VMAT planning for nasopharyngeal carcinoma, given adequate planning time.

## INTRODUCTION

Nasopharyngeal carcinoma (NPC) is a prevalent malignancy in Southeast Asia and southern China (*Chen et al., 2021*). Due to the biological characteristics of NPC and its anatomical location, radiotherapy has become the primary treatment modality (*Sun et al., 2019*; *Chen et al., 2019*). Intensity-modulated radiotherapy (IMRT), volumetric-modulated arc radiotherapy (VMAT), and hybrid IMRT/VMAT are widely used for NPC treatment (*Verbakel et al., 2009*; *Lu et al., 2012*; *Sun et al., 2013*, *2022*; *Yuen, Li & Mak, 2022*). VMAT, in particular, delivers highly conformal dose distributions while sparing organs at risk and offers shorter treatment times and lower monitor units (MUs), potentially reducing treatment-related toxicities (*Otto, 2008*; *Vanetti et al., 2009*; *He et al., 2020*).

In the Eclipse treatment planning system (TPS), VMAT plan optimization is performed using the photon optimization (PO) algorithm with multiresolution (MR) iterative optimization, which employs a voxel-based structural model and a uniform matrix for dose resolution (*Varian Medical Systems, 2017*; *Binny et al., 2020*). A new feature within the PO algorithm, the convergence mode (CM), automates the extension of optimization time, potentially improving plan quality by allowing more iterations and reducing the cost function. Previous research has demonstrated that the use of CM in radiotherapy planning for cancers such as prostate and breast can enhance plan quality by improving PTV homogeneity and reducing low-dose exposure to OARs (*Rossi & Boman, 2022*).

Despite its potential, the impact of CM on NPC VMAT planning has not been systematically investigated. This study aims to compare the effects of different CMs ('Off', 'On', and 'Extended') on key dosimetric metrics, plan complexity, MU values, and gamma pass rates for NPC radiotherapy. By doing so, we seek to provide guidance on the optimal CM settings for NPC treatment.

## MATERIALS AND METHODS

### Patient selection

This study included 21 patients with NPC treated at our hospital between January 2021 and December. Approval for the retrospective analysis of patient data was obtained from the ethics committee of The First Affiliated Hospital of Guangzhou Medical University (Approval No. ES-2023-008-01). Due to the retrospective nature of the study, the requirement for written informed consent was waived. All patients were staged according to the 8th edition of the American Joint Committee on Cancer (AJCC) Manual for Staging of Cancer (*Amin et al., 2017*). CT scans for each NPC patient were performed in the supine position with a slice thickness of 3 mm. The clinical characteristics of the patients are detailed in Table 1. Patients ages ranged from 30 to 67 years, with a mean age of 51.1 years and a median age of 51 years.

**Table 1 The characteristics of patients with nasopharyngeal carcinoma (*n* = 21).**

|  | Number | % |
| --- | --- | --- |
| Sex |  |  |
| Male | 6 | 28.6 |
| Female | 15 | 71.4 |
| T |  |  |
| 1 | 2 | 9.5 |
| 2 | 6 | 28.6 |
| 3 | 9 | 42.9 |
| 4 | 4 | 19.0 |
| N |  |  |
| 0 | 2 | 9.5 |
| 1 | 5 | 23.8 |
| 2 | 12 | 57.1 |
| 3 | 2 | 9.5 |
| M |  |  |
| 0 | 20 | 95.2 |
| 1 | 1 | 4.8 |
| Stages |  |  |
| I | 0 | 0.0 |
| II | 2 | 9.5 |
| III | 13 | 61.9 |
| IV | 6 | 28.6 |

## Target volume delineation and dose prescription

All CT images were transferred to the Eclipse 15.6 planning system for accurate contouring of target volumes and OARs. The target volumes (GTVp, GTVn, CTV1, and CTV2) were contoured according to the 2021 Chinese Society of Clinical Oncology (CSCO) guidelines (*Tang et al., 2021*) and the international guidelines (*Lee et al., 2018*) for target volumes delineation in NPC. The planning target volumes (PTVs), including PTV70, PTV66, PTV60, and PTV54, were generated from GTVp, GTVn, CTV1, and CTV2, respectively, with an additional 3 mm margin. The prescribed doses for the PTV70, PTV66, PTV60, and PTV54 were 70, 66, 60 and 54 Gy, respectively. All VMAT plans were created by using the simultaneous integrated boost (SIB) technique with 30 fractions. OARs were meticulously delineated and reviewed by certified oncologists.

## Treatment planning

Plans were generated using six MV FFF photon beams and modulated with a Millennium 120 multileaf collimator from VitalBeam (Varian Medical Systems, Palo Alto, California, USA) in Eclipse V15.6. For each patient, three VMAT plans were created, with CMs set to 'Off', 'On', and 'Extended', respectively, using the same optimization objectives. The plans were neither paused during optimization nor reoptimized after the final dose calculation.

All plans employed double full arcs irradiation, with collimator angles set at 15° and 345° to minimize the tongue-and-groove effect. The maximum dose rate was set to 1,400 MU/min for VMAT. Dose calculations were performed using the Acuros XB algorithm (AXB) with a 2.5 mm calculation grid. All three plans for each patient were normalized according to the institutional plan protocol after dose calculation.

## Plan evaluation parameters

The data obtained from the dose-volume histogram (DVH) for all plans were analyzed, focusing on specific parameters for plan comparisons.

### PTV coverage

To evaluate the coverage of the PTVs, several metrics were assessed: the dose received by 2% of the PTV (D2%), the dose received by 95% of the PTV (D95%), the conformity index (CI), and the homogeneity index (HI). The CI, as defined by *Paddick (2000)*, assesses the conformity of the target volume dose distribution.

$$CI = V_{t.ref}/V_t \times V_{t.ref}/V_{ref} \tag{1}$$

where $V_{t.ref}$ is the volume of target covered by the reference isodose, $V_t$ is the target volume, and $V_{ref}$ is the volume of the reference isodose. A high CI (ranging from 0 to 1) indicates high conformal dose delivery to the target. HI (*Kataria et al., 2012*) was defined as follows:

$$HI = (D2\% - D98\%)/D50\% \tag{2}$$

The HI evaluates the homogeneity of the target dose distribution. A lower HI value indicates better homogeneity. An HI of 0 represents the ideal homogeneity.

### Plan complexity

The average leaf pair opening (ALPO) and modulation complexity score for VMAT (MCSv) were used to assess the complexity of each plan. The ALPO is calculated as a weighted average of all leaf pair openings of every nonzero leaf pair opening at all control points (*Zygmanski & Kung, 2001*). The formula for ALPO is as follows:

$$\text{ALPO}_{MU} = \sum_{k=1}^{k=K} \frac{\sum_{i=1}^{i=I-1}\left[\sum_{j=1}^{j=J}(|X_R - X_L|)_{i,j}\right] \cdot MU_{i,i\pm1}}{\left(\sum_{i=1}^{i=I-1} MU_{i,i\pm1}\right) \cdot \sum_{j=1}^{j=J} j}/K \tag{3}$$

where K represents the number of total fields, I represents the total number of control points in each field, J represents the number of effective compressor pairs at each control point (the effective compressors are compressor pairs with distances greater than 0.5 mm), $X_R$ ($X_L$) are the right (left) leaf positions and $MU_{i,i\pm1}$ represents the MU between control point $i$ and control point $i \pm 1$. With increasing field aperture size, the ALPO value typically increases. Therefore, a large ALPO value indicates lower plan complexity.

The MCSv, used to evaluate the complexity of the MLC patterns, is calculated by an in-house program based on the leaf sequence variability (LSV) parameter and aperture

area variability (AAV) (*McNiven, Sharpe & Purdie, 2010*; *Masi et al., 2013*). The formula for MCSv is as follows:

$$MCSv = \sum_{i=1}^{I-1}\left[\frac{(AAV_{CPi} \pm AAV_{CPi\pm1})}{2} \times \frac{(LSV_{CPi} \pm LSV_{CPi\pm1})}{2} \times \frac{MU_{CPi,i\pm1}}{MU_{arc}}\right] \quad (4)$$

where $i$ is the ordinal number of the current control point, $MU_{CPi,i\pm1}$ is the number of MUs delivered between two successive control points, and $MU_{arc}$ is the $MU$ number of total arcs. The MCSv ranges from 0 to 1, with a lower value indicating greater complexity.

### Organs at risk

For patients with NPC, the following values were determined: maximum doses (Dmax) to the brainstem, spinal cord, optic nerves, optic chiasm, lens, pituitary, mandibles, and TMJ; mean doses (Dmean) to the brainstem, spinal cord, eyes, parotid glands, larynx, oral cavity, and middle ears; volume receiving 30 Gy (V30) for the parotid glands; and the dose to 2% of the OAR volume (D2%) for the temporal lobes and mandibles.

### Plan optimization time

Total optimization time and the optimization time of each MR level for each plan were determined and compared.

## Quality assurance

Plan-specific QA measurements were performed using the Varian VitalBeam Portal Dosimetry system for all VMAT plans. Gamma analysis assessed the agreement between measured and calculated dose distributions, with threshold settings of 10% at 3%/3 and 2%/2 mm global passing rates.

## Statistical analysis

Statistical analysis was conducted using SPSS 25.0 (SPSS, Inc., Chicago, IL, USA). Results are presented as mean ± standard deviation. Wilcoxon's signed rank test was used to analyze the data from three different treatment plans, with $p < 0.05$ considered statistically significant.

## RESULTS

### Dosimetry for PTVs and OARs

All parameters obtained from the three types of plans for NPC complied with the ICRU 83 recommendation for target coverage. Overall, all plans based on different CMs met clinical prescription requirements. The dose parameters for the PTVs are shown in Table 2. For all PTVs, the HI and D2% were lower with CM 'On' or 'Extended' compared to CM 'off'. Furthermore, for the four PTVs, plans with CM 'On' or 'Extended' exhibited higher CI values than those with CM 'Off' (***$p < 0.001$). When comparing CMs 'On' and 'Extended', more than half of the parameters showed significant differences (HI and CI of PTV70; D2% and CI of PTV66; D2%, D95% and CI of PTV60; D2% and CI of PTV54). Among the three types of plans, CM 'Extended' demonstrated superior HI and CI values

**Table 2 Dosimetric comparison of PTVs in three VMAT plans (mean ± SD).**

| Targets | Index | Off | On | Extended | p | | |
|---|---|---|---|---|---|---|---|
| | | | | | On *vs.* Off | Extended *vs.* Off | Extended *vs.* On |
| PTV70 | D2% | 75.19 ± 0.84 | 74.88 ± 0.74 | 74.79 ± 0.62 | **<0.001** | **<0.001** | 0.061 |
| | D95% | 70.27 ± 0.42 | 70.23 ± 0.33 | 70.21 ± 0.3 | 0.061 | 0.123 | 0.948 |
| | HI | 0.080 ± 0.015 | 0.077 ± 0.015 | 0.075 ± 0.014 | **<0.001** | **<0.001** | **0.039** |
| | CI | 0.682 ± 0.095 | 0.721 ± 0.100 | 0.741 ± 0.090 | **<0.001** | **<0.001** | **0.007** |
| | Volume | | | 65.83 ± 41.81 cm$^3$ | | | |
| PTV66 | D2% | 71.8 ± 1.04 | 71.29 ± 0.95 | 71.14 ± 0.89 | **<0.001** | **<0.001** | **0.042** |
| | D95% | 66.48 ± 0.65 | 66.41 ± 0.56 | 66.32 ± 0.57 | 0.118 | **0.019** | 0.117 |
| | CI | 0.248 ± 0.115 | 0.255 ± 0.117 | 0.258 ± 0.119 | **<0.001** | **<0.001** | **0.002** |
| | HI | 0.090 ± 0.016 | 0.083 ± 0.014 | 0.083 ± 0.014 | **<0.001** | **<0.001** | 0.59 |
| | Volume | | | 50.18 ± 40.02 cm$^3$ | | | |
| PTV60 | D2% | 74.78 ± 0.88 | 74.47 ± 0.77 | 74.35 ± 0.65 | **<0.001** | **<0.001** | **0.037** |
| | D95% | 61.33 ± 0.85 | 61.19 ± 0.77 | 61.07 ± 0.73 | **0.001** | **0.001** | **0.019** |
| | CI | 0.495 ± 0.139 | 0.523 ± 0.141 | 0.542 ± 0.146 | **<0.001** | **<0.001** | **<0.001** |
| | HI | 0.217 ± 0.018 | 0.213 ± 0.016 | 0.213 ± 0.014 | **0.001** | **0.005** | 0.664 |
| | Volume | | | 196.05 ± 121.62 cm$^3$ | | | |
| PTV54 | D2% | 73.86 ± 1.02 | 73.58 ± 0.95 | 72.24 ± 5.61 | **<0.001** | **<0.001** | **0.022** |
| | D95% | 53.27 ± 0.74 | 53.41 ± 0.65 | 53.43 ± 0.63 | **0.01** | **0.039** | 0.394 |
| | HI | 0.384 ± 0.035 | 0.377 ± 0.033 | 0.375 ± 0.032 | **<0.001** | **0.001** | 0.217 |
| | CI | 0.843 ± 0.018 | 0.851 ± 0.016 | 0.856 ± 0.015 | **<0.001** | **<0.001** | **<0.001** |
| | Volume | | | 748.45 ± 177.51 cm$^3$ | | | |

**Note:**
Bold *p* values indicate statistical significance ($p < 0.05$); PTV, planning target volume; D2%, dose covering 2% of the target volume; D95%, dose covering 95% of the target volume; HI, homogeneity index; CI, conformity index.

for PTV70 and PTV54. Typical dose–volume histograms for the three VMAT plans are presented in Fig. 1.

The dosimetric parameters for all OARs are summarized in Table 3. The OARs received the lowest doses with CM 'Extended', followed by CM 'On' and CM 'Off'. The Dmax values for the spinal cord and optic nerve right were significantly lower with CM 'On' compared to CM 'Off' (*$p \leq 0.05$). Additionally, Dmean values for the brainstem, spinal cord, bilateral parotids, larynx, oral cavity, bilateral ears, and V30 for the left parotid in plans with CM 'On' were also lower than those with CM 'Off' (*$p \leq 0.05$). When comparing CM 'Extended' and CM 'Off', more parameters showed statistically significant differences (Dmax values for the brainstem, spinal cord, right optic nerve, optic chiasm, bilateral lens, and pituitary; Dmean values for the brainstem, spinal cord, bilateral eyes, bilateral parotids, larynx, oral cavity, and bilateral middle ears; V30 of bilateral parotids; D2% of right temporal lobe and mandibles). Except for Dmax of the left optic nerve, bilateral TMJ, and D2% of the left temporal lobe, which were not significantly different, the remaining 21 parameters had significantly lower values with CM 'Extended'. The dose indices for the spinal cord, bilateral lens, right eye, bilateral parotids, right temporal lobe,

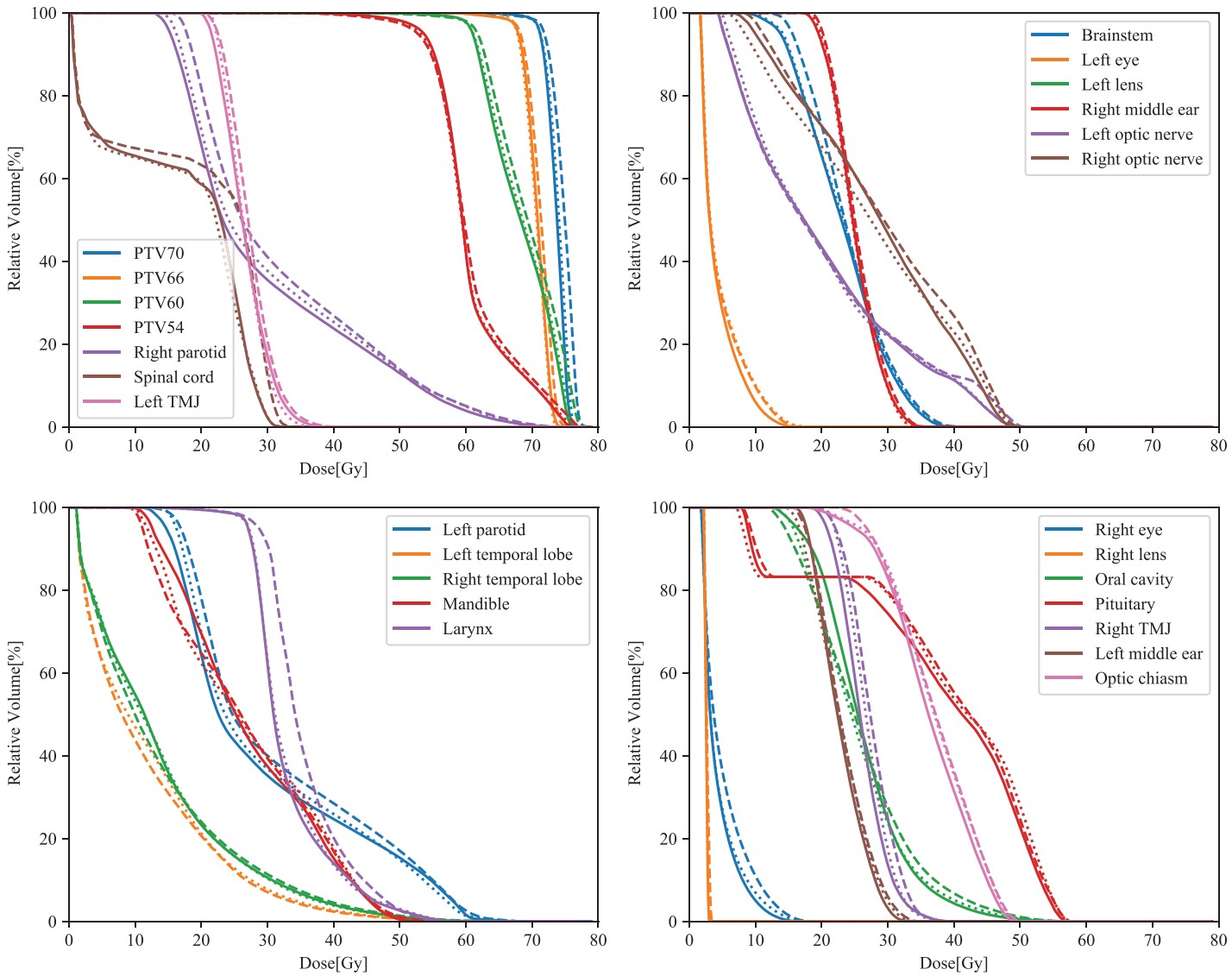

**Figure 1** Typical DVH curves of the 'Off' (dashed lines), 'On' (dotted lines), and 'Extended' (solid lines) plans in NPC.

mandibles, larynx, oral cavity, and right middle ear were lower with CM 'Extended' compared to CM 'On' (*$p \leq 0.05$).

## Plan complexity

The results of plan complexity for the three plans are shown in Figs. 2A and 2B. The 'Extended' CM achieved the lowest ALPO value of 28.4 ± 4.2 mm, significantly lower than that of the 'Off' CM (29.1 ± 4.3 mm) (**$p = 0.004$). Additionally, the ALPO value for the 'On' CM was 28.6 ± 4.2 mm, which was also significantly lower than that for the 'Off' CM (**$p = 0.002$). While the ALPO for 'Extended' was slightly lower than that for 'On', the difference was not statistically significant. A similar trend was observed for MCSv. The MCSv values for the CM 'Off', 'On', and 'Extended' groups were 0.1730 ± 0.0215, 0.1691 ±

**Table 3 Dosimetric comparison of OARs in three VMAT plans (mean ± SD).**

| OARs | Index | Off | ON | Extended | p | | |
|---|---|---|---|---|---|---|---|
| | | | | | On vs Off | Extended vs Off | Extended vs On |
| Brainstem | Dmax(Gy) | 50.91 ± 6.47 | 50.69 ± 6.62 | 50.33 ± 6.46 | 0.099 | **0.013** | 0.054 |
| | Dmean(Gy) | 27.57 ± 4.6 | 27.11 ± 4.51 | 26.94 ± 4.55 | **0.006** | **0.001** | 0.063 |
| Spinal cord | Dmax(Gy) | 39.58 ± 2.92 | 38.59 ± 2.63 | 37.72 ± 2.97 | **0.001** | **<0.001** | **0.001** |
| | Dmean(Gy) | 23.34 ± 2.21 | 22.74 ± 2.26 | 22.08 ± 2.26 | **<0.001** | **<0.001** | **<0.001** |
| Left optic nerve | Dmax(Gy) | 41.93 ± 14.38 | 41.74 ± 14.21 | 41.62 ± 14.21 | 0.192 | 0.159 | 0.375 |
| Right optic nerve | Dmax(Gy) | 47.55 ± 12.68 | 47.14 ± 12.75 | 47.02 ± 12.62 | **0.004** | **0.009** | 0.664 |
| Optic chiasm | Dmax(Gy) | 44.88 ± 12.75 | 44.66 ± 12.83 | 44.46 ± 12.89 | 0.068 | **0.033** | 0.23 |
| Left lens | Dmax(Gy) | 3.4 ± 1.2 | 3.32 ± 1.08 | 3.21 ± 1.02 | 0.498 | **0.002** | **0.017** |
| Right lens | Dmax(Gy) | 3.54 ± 0.92 | 3.53 ± 0.8 | 3.38 ± 0.77 | 0.876 | **0.046** | **0.004** |
| Left eye | Dmean(Gy) | 5.59 ± 3.13 | 5.41 ± 2.82 | 5.39 ± 2.89 | 0.274 | **0.011** | 0.296 |
| Right eye | Dmean(Gy) | 5.98 ± 2.69 | 5.9 ± 2.6 | 5.79 ± 2.48 | 0.375 | **0.035** | **0.039** |
| Left parotid | Dmean(Gy) | 27.62 ± 2.99 | 27.16 ± 2.83 | 26.75 ± 2.75 | **<0.001** | **<0.001** | **<0.001** |
| | V30(%) | 35.0 ± 7.7 | 34.1 ± 7.4 | 33.4 ± 7.1 | **0.003** | **<0.001** | **0.002** |
| Right parotid | Dmean(Gy) | 26.14 ± 3.57 | 25.62 ± 3.47 | 25.23 ± 3.38 | **<0.001** | **<0.001** | **<0.001** |
| | V30(%) | 31.2 ± 7.7 | 30.9 ± 7.3 | 30.2 ± 7.0 | 0.161 | **0.012** | **0.009** |
| Left temporal lobe | D2%(Gy) | 48.41 ± 8.86 | 48.13 ± 8.8 | 48.08 ± 8.44 | 0.192 | 0.274 | 0.931 |
| Right temporal lobe | D2%(Gy) | 48.82 ± 7.73 | 48.76 ± 7.55 | 48.3 ± 7.59 | 0.821 | **0.017** | **0.035** |
| Mandible | D2%(Gy) | 55.02 ± 4.34 | 54.75 ± 4.6 | 54.49 ± 4.58 | 0.159 | **0.011** | **0.039** |
| Larynx | Dmean(Gy) | 35.33 ± 4.35 | 34.59 ± 4.33 | 34.23 ± 4.06 | **<0.001** | **<0.001** | **0.002** |
| Oral cavity | Dmean(Gy) | 33.07 ± 5.78 | 32.2 ± 5.56 | 31.85 ± 5.55 | **<0.001** | **<0.001** | **0.025** |
| Pituitary | Dmax(Gy) | 58.59 ± 9.66 | 58.46 ± 9.22 | 58.17 ± 9.1 | 0.23 | **0.017** | 0.054 |
| Left TMJ | Dmax(Gy) | 42.24 ± 9.12 | 41.78 ± 9.26 | 42.01 ± 8.92 | 0.149 | 0.305 | 0.33 |
| Right TMJ | Dmax(Gy) | 46.8 ± 9.14 | 47.12 ± 9.41 | 46.85 ± 9.39 | 0.274 | 0.876 | 0.305 |
| Left middle ear | Dmean(Gy) | 34.92 ± 7.65 | 34.38 ± 7.22 | 34.16 ± 7.22 | **0.002** | **0.001** | 0.052 |
| Right middle ear | Dmean(Gy) | 35.74 ± 8.26 | 35.4 ± 8.2 | 35.14 ± 8.17 | **0.002** | **<0.001** | **0.006** |

Note:
Bold p values indicate statistical significance (p < 0.05); Dmax, maximum dose; Dmean, mean dose; V30, Relative volume covered by 30Gy; D2%, Dose covering 2% of the target volume.

0.0204, and 0.1693 ± 0.0208, respectively. The MCSv values for CM 'On' (***$p < 0.001$) and 'Extended' (**$p = 0.006$) were significantly lower than those for CM 'Off'. The MCSv was slightly lower in the CM 'Extended' group than in the CM 'On' group, but this difference was not statistically significant.

## Monitor unit and plan optimization time

The average MUs were 796.2 ± 110.8 for CM 'Off', 798.6 ± 106.1 for CM 'On' and 799.7 ± 103.6 for CM 'Extended'. The MUs were similar among the three VMAT plans ($p > 0.05$), as shown in Fig. 2C. To assess planning efficiency, the plan optimization times for the three VMAT plans were analyzed. As shown in Fig. 3A, the mean total optimization times for CM 'Off', 'On', and 'Extended' were 14.4 ± 3.2, 35.9 ± 8.9, and 145.6 ± 50 min, respectively. The total optimization times differed significantly between the CMs (***$p < 0.001$).

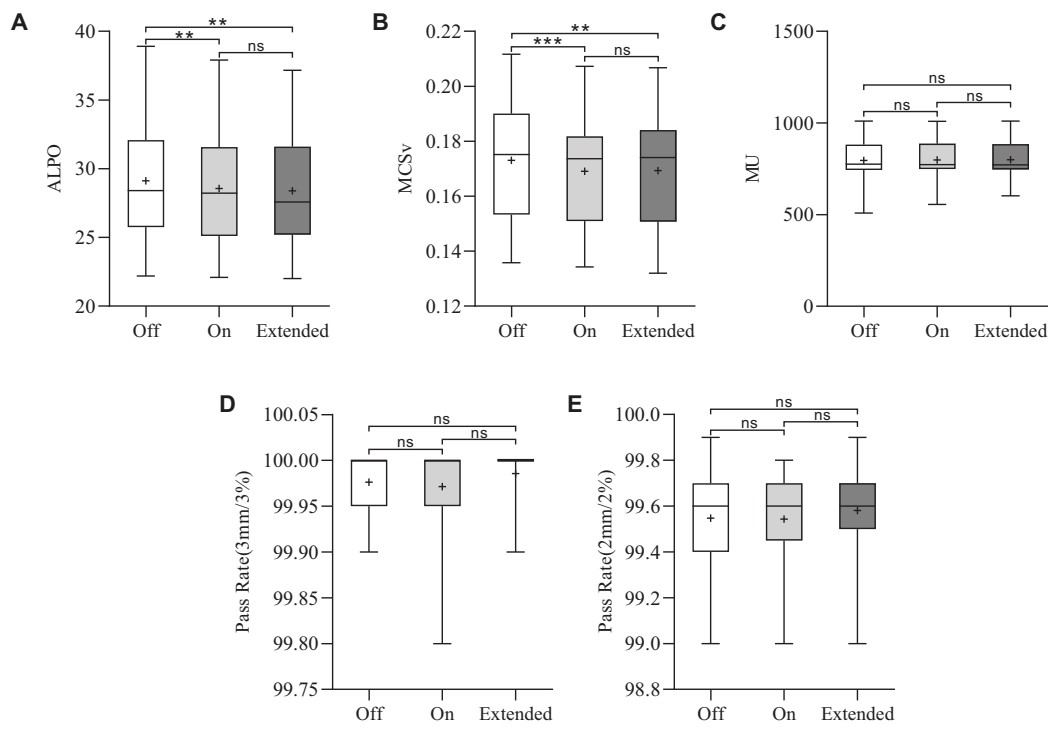

**Figure 2 Statistical comparison of the MCSv, ALPO, MUs, and pass rates for the three types of plans.**
*$p < 0.05$, **$p < 0.01$, ***$p < 0.001$.   

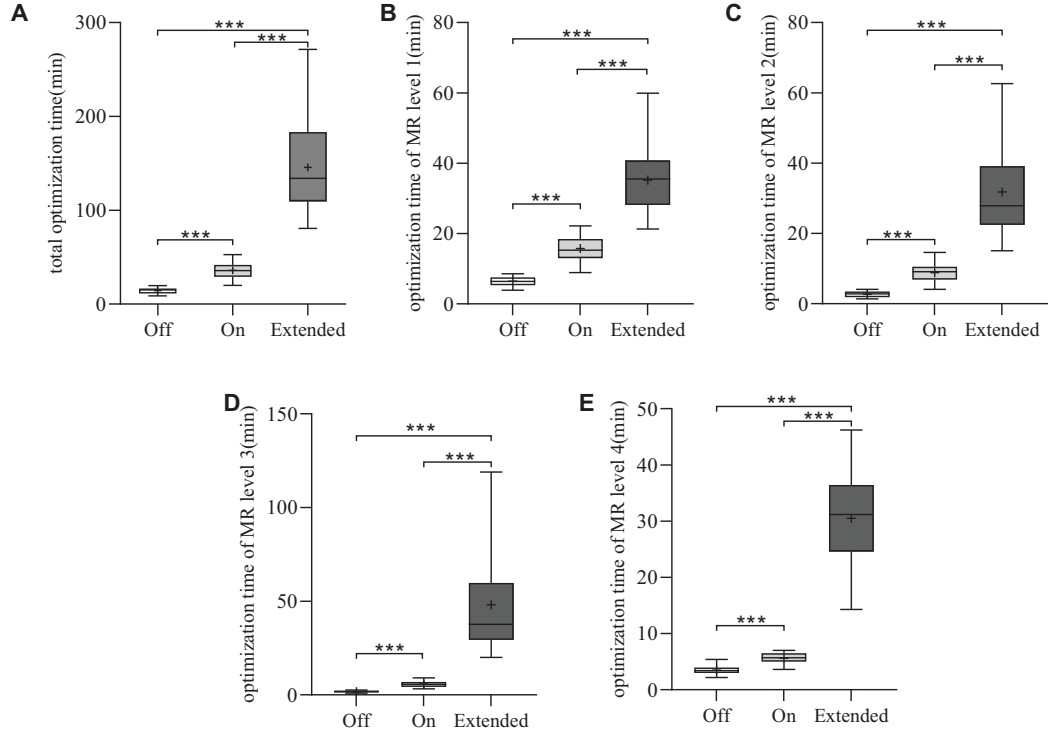

**Figure 3 Statistical comparison of the optimization time for the three types of plans.** *$p < 0.05$,
**$p < 0.01$, ***$p < 0.001$.   

Additionally, as shown in Fig. 3B to 3E, significant differences in optimization times were observed at each MR level among the three CMs (***$p < 0.001$).

## Quality assurance

For all three VMAT plans, the measured dose distributions closely matched the calculated dose distributions. The gamma pass rates ranged from 99.8% to 100% (3%/3 mm criterion) and 99% to 99.9% (2%/2 mm criterion), as shown in Figs. 2D and 2E. Overall, no significant differences were observed among the different CMs.

## DISCUSSION

Designing VMAT plans for nasopharyngeal carcinoma patients is challenging due to the complex anatomical structures involved. The quality of a VMAT plan largely depends on the planner's expertise and the appropriate setting of TPS parameters. Variations in VMAT planning factors, such as optimization algorithms, number of arcs, collimator angle, grid size, optimization objectives, aperture shape controller, and convergence modes, significantly impact the final dose distribution (*Laura Licon et al., 2018*; *Jihong et al., 2022*).

The Eclipse VMAT optimization process utilizes the PO algorithm with multiresolution iterative optimization. A relatively new feature in PO optimization, CM, allows users to select among 'Off', 'On', or 'Extended'. To the best of our knowledge, the impact of different CMs on VMAT planning for NPC has not been systematically studied. Therefore, this study aimed to evaluate the influence of various CMs on VMAT planning for NPC patients, with a focus on identifying the most effective CM. The study compared VMAT plans generated using different CMs based on parameters including PTVs, OARs, CI, HI, MCSv, ALPO, MUs, optimization time, and gamma pass rates.

Our findings revealed that all three CMs met clinical dosimetry requirements for NPC treatment using VMAT. Notably, the 'On' and 'Extended' CMs improved PTV conformity and homogeneity compared to the 'Off' mode. CI values for all PTVs were higher with 'On' and 'Extended' CMs, with the 'Extended' mode yielding the highest CI. HI values were also lower with these CMs, indicating better homogeneity, particularly in PTV70 for the 'Extended' mode. Our results align with those of *Rossi & Boman (2022)*, who found that CM 'On' improved HI, V95%, and D2cc in prostate, breast, and head & neck cancer plans. However, their study did not report significant improvements in CI with CM 'On', while our study demonstrated that both CM 'On' and 'Extended' significantly improved CI and HI in NPC plans. These discrepancies highlight the complexity of VMAT planning, as the effects of CM may vary depending on the treatment site and complexity of the case. Additionally, they suggested avoiding CM 'Extended' in simpler cases, which contrasts with our finding that all NPC cases, regardless of complexity, benefitted from the use of CM 'Extended'.

In terms of OAR sparing, the 'Extended' mode consistently provided the best sparing, followed by 'On' and 'Off'. A total of 25 dosimetric indices for the OARs were compared, including 10 Dmax, 10 Dmean, three D2%, and two V30 indices. In the CM 'On' mode, 11 indices showed significantly lower values compared to the 'Off' mode, with eight of these

being Dmean. The 'Extended' CM mode demonstrated significantly lower values in 21 indices compared to the 'Off' mode, with all Dmean indices showing statistically significant differences. When comparing the 'Extended' and 'On' modes, 14 indices exhibited lower values, including seven Dmean indices. This reduction in OAR doses, particularly in Dmean values, indicates that the extended optimization time in these modes allows for more refined leaf sequencing and improved dose distribution. This is particularly relevant for critical OARs, where small reductions in dose can significantly affect patient outcomes. These findings are consistent with the results of *Davis et al. (2024)*, who observed improved brain sparing and PTV coverage using CM 'Extended' in total scalp irradiation.

Plan complexity was assessed using ALPO, MCSv, and MUs. ALPO, which measures plan sensitivity to leaf position errors, and MCSv, which reflects modulation, were both lower for the 'On' and 'Extended' modes, indicating increased plan complexity. No significant differences in ALPO or MCSv were found between the two modes. The 'Extended' CM yielded the lowest ALPO value. The CM allows the optimizer to run for a longer duration, minimizing the cost function. This improves dose distribution and overall plan quality, but also increases optimization time and may raise plan complexity. However, MUs did not significantly differ among the three CM modes. These results suggest that extended optimization time enhances dose distribution but may introduce complexity without improving delivery efficiency.

For the four MR levels during optimization, the maximum number of iterations increased by 2.5/11.2 times for MR1, 2/17.8 times for MR2, 1/17 times for MR3, and 1/15 times for MR4 in the 'On' and 'Extended' modes, respectively (*Varian Medical Systems, 2017*). Naturally, a higher number of iterations results in longer optimization times. The increase in optimization time with CMs was substantial, with CM 'On' taking 2.5 times longer than CM 'Off' and CM 'Extended' taking up to ten times longer. Our study showed that for MR levels 1–4, the average optimization times for CM 'On' were 2.5, 3.3, 3.4, and 1.6 times those of the 'Off' mode, respectively. For CM 'Extended', the average optimization times for MR levels 1–4 were 5.5, 11.8, 28.3, and 8.7 times those of the 'Off' mode. Although this extended optimization time could pose a barrier to clinical implementation, potential solutions include running optimizations during off-hours or using GPUs to accelerate computation. *Rossi & Boman (2022)* reported similar increases in optimization time, particularly at the third MR level, which aligns with our findings. Despite these variations in optimization time, no significant differences in gamma pass rates were observed between the three CMs when evaluated using the 3%/3 or 2%/2 mm criteria. This suggests that the choice of CM does not affect the deliverability of the plan, as all plans achieved high gamma pass rates, confirming their feasibility for clinical delivery.

*Unkelbach et al. (2015)* suggested that VMAT optimization may yield local rather than global optima, potentially resulting in variations in results even under identical optimization conditions. To evaluate this effect, we conducted a case study by generating five VMAT plans for each CM setting ('Off', 'On', and 'Extended') using identical optimization objectives. The analysis revealed that although the outcomes from the repeated optimizations were not always identical, the observed variations were minor and substantially smaller than the differences associated with different CM settings. For

example, the mean V30 values for bilateral parotids across five repeated optimizations were 39.7%, 37.2%, and 34.8% for 'Off', 'On', and 'Extended', respectively, demonstrating statistically significant differences between the CM settings. Similar trends were observed for target volumes and other organs at risk (OARs), further supporting the consistency of the findings. These findings suggest that the slight variations arising from repeated optimizations have a negligible impact on the validity of our conclusions.

While this study provides detailed information about the effects of CMs on VMAT planning for NPC, it has several limitations. The study was conducted at a single center, and the results are based on a limited sample of 21 patients. Additionally, variations in TPS configurations and clinical protocols may influence the generalizability of our findings. Future work should include a larger cohort and investigate other tumor sites to further validate these results.

## CONCLUSIONS

All plans generated using the three convergence modes can produce dose distributions that meet clinical requirements. The use of CMs ('On' and 'Extended') improves the conformity and homogeneity of the PTVs and reduces the radiation dose to OARs, particularly in terms of the mean dose, without compromising target coverage. Among the CMs, the 'Extended' mode demonstrates the most significant effect. Although CMs slightly increase plan complexity, as evidenced by reduced MSCv and ALPO values, they do not affect the number of MUs or gamma pass rates. While optimization time is prolonged for both the 'On' and 'Extended' modes, it remains within acceptable limits. Therefore, the 'On' or 'Extended' CM may be used when designing VMAT plans for nasopharyngeal cancer, as it improves target volume conformity and homogeneity while improving OAR sparing.

### Funding

This work was supported by the Medical Science and Technology Foundation of Guangdong Province (Grant No. A2023250 and B2023442), the Science and Technology Program of Guangzhou (202201020436), and the Project of the Featured Clinical Technique of Guangzhou (2019TS28). The funders had no role in study design, data collection and analysis, decision to publish, or preparation of the manuscript.

### Grant Disclosures

The following grant information was disclosed by the authors:
Medical Science and Technology Foundation of Guangdong Province: A2023250 and B2023442.
Science and Technology Program of Guangzhou: 202201020436.
Featured Clinical Technique of Guangzhou: 2019TS28.

## Competing Interests

The authors declare that they have no competing interests.

## Author Contributions

- Maoying Lan performed the experiments, analyzed the data, prepared figures and/or tables, authored or reviewed drafts of the article, and approved the final draft.
- Rui Wu performed the experiments, prepared figures and/or tables, and approved the final draft.
- Guanhua Deng analyzed the data, prepared figures and/or tables, authored or reviewed drafts of the article, and approved the final draft.
- Bo Yang performed the experiments, prepared figures and/or tables, and approved the final draft.
- Yongdong Zhuang analyzed the data, authored or reviewed drafts of the article, and approved the final draft.
- Wei Yi conceived and designed the experiments, authored or reviewed drafts of the article, and approved the final draft.
- Wenwei Xu conceived and designed the experiments, performed the experiments, prepared figures and/or tables, and approved the final draft.
- Jiancong Sun conceived and designed the experiments, authored or reviewed drafts of the article, and approved the final draft.

## Human Ethics

The following information was supplied relating to ethical approvals (*i.e.*, approving body and any reference numbers):

The Institutional Review Board of the First Affiliated Hospital of Guangzhou Medical University (Approval Number: ES-2023-008-01).

## Data Availability

The raw measurements are available in the Supplemental Files.

## Supplemental Information

Supplemental information for this article can be found online at http://dx.doi.org/10.7717/peerj.18773#supplemental-information.

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
