# Peer review of "Dosimetric comparison and evaluation of different convergence modes in nasopharyngeal carcinoma using VMAT treatment deliveries"

_PeerJ, doi:10.7717/peerj.18773_

## Round 0.1 · original submission · Major Revisions

The authors are requested to carefully revise the manuscript and answer the questions raised by the reviewers.

·

Basic reporting

The authors have written an interesting article. The English writing is sound. I would suggest the authors to consider the comments below before publication:

Introduction

1. Line 59: Typo in 'China'

2. In the begining of this manuscript, the authors have mentioned that 'Both IMRT and VMAT are widely used standard radiotherapy methods for NPC'. In fact, currently there are alot of novel radiotherapy methodology, such as "Hybrid-VMAT", "Hybrid IMRT". Therefore, before jumping into the main content of this paper, I think the authors should use a paragraph to introduce the "History" of radiotherapy techniques in NPC (e.g. 3D>IMRT>VMAT>Hybrid-VMAT etc.) and why only VMAT is used in this study, but not hybrid VMAT / IMRT etc.

3. Line 74: The term 'PO' should not be abbreviation in its first appearance (i.e. photon optimization (PO) algorithm)

4. Line 79 (physicists can utilize the hold function at each step of each level): I think it is not proper to use physicists as the subject here, since in current radiotherapy practice, professionals such as oncologists, radiation therapists / technicians could also be an user of the TPS. I would prefer the author to use the term 'user' instead

5. Line 82-83: It is appreciative that the author has mentioned a longstanding 'myth' in the radiotherapy industry - (i.e. the use of the hold function at each step necessitates human intervention,and the determination of the appropriate holding time remains a question), but the authors didn't include manual based optimization in this study as one of the study arm (or 'off' mode is manual optimization?). In this case, I think the author should clarify this part in methodology. Also, I think the author should record the time used in each MR level and steps, instead of the overall time in each study arm. This is important since that holding 500 seconds in MR level 4 is not equivalent to holding 500 seconds in MR level 1.


Methodology

7. Line 112: year of reference (i.e. Lee et al.,) is not the same in reference. 2018 or 2018

8. line 183: p should be italicized

9. The optimization objectives of the plans should be listed


Results

10. Line 192: (more than half of the items) which items? they should be listed clearly. That could be too vague to say in this way

11. Line 203: (more items showed statistically significant differences) see comments above



Discussion

12. I think in the first place of discussion, the significance of the results of the present study should be mentioned, rather than Line 236-246

13. Line 236-246: These should be something exist in introduction rather than in discussion

14. Line 247: the authors mentioned that "The design of the VMAT plan for nasopharyngeal carcinoma patients is complicated" however, only 2 arcs were used for planning, I think this needed to be further discussed (3 arcs were generally used in many studies)

15. Line 254-255: This flexibility could be a subjective variation, so this should also be recorded in methodology (see comment 5)

16. Line 257-274: These paragraphs should be the first and second paragraph of discussion

17. I think it is interesting to study different convergence mode, it would be better if the authors could discuss the results of other researchers (or other primary cancer site) who also adopted this methods

Experimental design

no comment

Validity of the findings

no comment

Reviewer 2 ·

Basic reporting

This paper compared different convergence mode for in nasopharyngeal carcinoma VMAT planning, which has certain practicality and reference value for radiotherapy physicists. The author should carefully review the writing of the article and revise it throughout to make it more coherent and readable.
But I believe this study needs to be rewritten. Simplify the introduction section, and concisely state the research background and significance. The research methods need to be adjusted. The discussion section needs more in-depth discussion

Experimental design

The methods should be described more clearly. And some questions should be made clear, which were attached in the file.

Validity of the findings

There is some meaningful results, but the discussion should be rewritten. The discussion should not be another introduction and the explanation of results. Some in-depth discussion should be included.

Additional comments

This article should be rewritten.

Annotated reviews are not available for download in order to protect the identity of reviewers who chose to remain anonymous.

Reviewer 3 ·

Basic reporting

This study investigates the use of Convergence Mode (CM) in the Eclipse treatment planning system for volumetric modulated arc therapy (VMAT) in nasopharyngeal carcinoma (NPC). The study focuses on plan quality, complexity, and dosimetric verification using different CM settings (Off, On, and Extended).

Basic Reporting
• Language and Presentation: The manuscript is written in clear and professional English. The introduction effectively sets the context and provides a comprehensive background, situating the study within the existing body of literature.
• Figures and Tables: The figures and tables are relevant, of high quality, and are appropriately labeled and described.
• References: The literature is well-cited, up-to-date, and supports the scientific context of the study.

Experimental design

Experimental Design
• Research Question: The research question is well-defined, relevant, and meaningful. It clearly addresses an identified knowledge gap in the field.
• Methods: The methods are described with sufficient detail and information to allow replication. The study adheres to high technical and ethical standards.

Validity of the findings

Validity of the Findings
• Data Robustness: All underlying data are provided, robust, statistically sound, and controlled.
• Conclusions: The conclusions are well-stated, linked to the original research question, and supported by the results.

Additional comments

Strengths
1. Plan Quality and Complexity: The detailed analysis of the impact of different CM settings on plan quality and complexity provides valuable insights for clinical applications.
2. Dosimetric Verification: The use of gamma pass rates for dosimetric verification demonstrates the clinical applicability of the plans.
3. Ethical Approval: The study received ethical approval and adhered to all necessary ethical standards.
Weaknesses and Recommendations for Improvement
1. Sample Size: The sample size of 21 patients may limit the generalizability of the findings. Repeating the study with a larger sample size could strengthen the conclusions.
2. Optimization Time: The optimization time for the Extended CM setting is notably long, which could present practical challenges. Including a discussion on potential solutions or workarounds for this issue would be beneficial.
Questions
1. Gamma Pass Rates: Is there a statistically significant difference in gamma pass rates between the various CM modes?
2. Plan Complexity and Deliverability: What are the effects of different CM modes on plan complexity and deliverability?
3. Convergence Modes and OARs: How do the convergence modes impact the doses to organs at risk (OARs)?
4. Plan Optimization Time: What are the differences in plan optimization time between the various convergence modes?
5. Dose Homogeneity and Conformity Index: What are the effects of different convergence modes on dose uniformity (HI) and conformity index (CI)?
6. Dosimetry Parameters: What are the effects of different CM modes on the dosimetry parameters determined for PTV and OARs?
7. Modulation Complexity and Leaf Pair Opening: What are the effects of CM modes on modulation complexity (MCSv) and average leaf pair opening (ALPO)?
8. Monitor Units (MU): What are the effects of different CM modes on total monitor units (MU)?

---

## Round 0.2 · Major Revisions

The authors are requested to carefully revise the manuscript and respond to the comments of Reviewer 2.

Reviewer 2 ·

Basic reporting

The manuscript should be rewritten. The article should be clear and technically correct.

Experimental design

Research question well defined, relevant & meaningful. It is stated how research fills an identified knowledge gap.

Validity of the findings

I believe this study needs to be rewritten. Simplify the introduction section, and concisely state the research background and significance. The research methods need to be adjusted. The discussion section needs more in-depth discussion.

Additional comments

The introduction is too long, especially the first paragraph, so there is no need to use so much text to state that VMAT is an important and most commonly used technique in NPC radiotherapy planning.

The pause function (Line 81, Line 250) is the same with hold function.

The target volumes (GTVp, GTVn, CTV1, and CTV2) were delineated according to an international guideline (Lee et al., 2018) by experienced radiation oncologists specializing in head and neck radiotherapy (Line 108-110). What international guideline did you follow to name the targets?

“The prescription doses for the PTVs were 70 Gy for GTVp, 66 Gy for GTVn, 60 Gy for CTV1, and 54 Gy for CTV2 (Line 110-112).” The prescription of targets should be clear in this part.

PTV coverage. The D95% of PTVs were not evaluated in this study?

“D2% of right temporal lobe and mandible (Line 206-207)” Why the right temporal lobe and mandible were evaluated, or the bilateral temporal lobes and mandibles should be evaluated. The “left eye, right eye, left parotid, right parotid, larynx, oral cavity, left middle ear, and right middle ear; V30 of left and right parotids” should be bilateral eyes, bilateral parotids, bilateral middle ears (L204-208)...

“The results of plan complexity for the three plans are shown in Fig. 2A and Fig. 2B. The ‘Extended’ CM achieved the lowest ALPO value of 28.4±4.2 mm, ” the P value could be added in the text (Line 213-220).

p = 0.004 should be **p = 0.004. Please revise all the P value in the manuscript.

Discussion

The first paragraph was too much about the introduction of IMRT or VMAT, however, there is no need to illustrate these. Just make your research significance more clear with simple words.

Annotated reviews are not available for download in order to protect the identity of reviewers who chose to remain anonymous.

Reviewer 3 ·

Basic reporting

• Language and Presentation: The manuscript is written in clear and professional English. The introduction effectively sets the context and provides a comprehensive background, situating the study within the existing body of literature.
• Figures and Tables: The figures and tables are relevant, of high quality, and are appropriately labeled and described.
• References: The literature is well-cited, up-to-date, and supports the scientific context of the study.

Experimental design

• Research Question: The research question is well-defined, relevant, and meaningful. It clearly addresses an identified knowledge gap in the field.
• Methods: The methods are described with sufficient detail and information to allow replication. The study adheres to high technical and ethical standards.

Validity of the findings

• Data Robustness: All underlying data are provided, robust, statistically sound, and controlled.
• Conclusions: The conclusions are well-stated, linked to the original research question, and supported by the results.

Additional comments

This study investigates the use of Convergence Mode (CM) in the Eclipse treatment planning system for volumetric modulated arc therapy (VMAT) in nasopharyngeal carcinoma (NPC). The study focuses on plan quality, complexity, and dosimetric verification using different CM settings (Off, On, and Extended).
The queries I posed were duly addressed, and the pertinent sections were duly amended by the requisite corrections.

---

## Round 0.3 · Minor Revisions

The authors are requested to carefully revise the manuscript and answer the questions raised by the reviewers.

·

Basic reporting

The topic is interesting, but solution is not obvious. Authors use very rudimentary approach for plan variation. In Eclipse there are many parameters but most of them have been looked for example Aperture shape (Binny et al, Medical Dosimetry 45 (2020) 284–292). The most complex problem authors failed to show is repeated planning and its convergence as shown by Unkelbach et al 1367 Med. Phys. 42 (3), 2015.
As such this is well written but its impact on clinical practice is minimum.

Experimental design

Ok except as I said above. It might be interesting if you can pick a select cases and do planning say 5 times and see the variation.

Validity of the findings

Ok and has no impact on clinical practice.

Additional comments

None

---

## Round 0.4 · accepted · Accept

After revisions, two reviewers agreed to publish the manuscript. I also reviewed the manuscript and found no obvious risks to publication. Therefore, I also approve the publication of this manuscript.

Reviewer 2 ·

Basic reporting

The topic was clearly expressed with sufficient field background and literature references.

Experimental design

The research was well defined and described with sufficient information.

Validity of the findings

It is a a meaningful research to explore how the convergence modes affect the optimization for NPC plans.

Additional comments

The language may need to be polished.